# Prevalence and Clinical Profile of Adults with ADHD Attending a Tertiary Care Hospital for Five Years

**DOI:** 10.3390/ijerph21050566

**Published:** 2024-04-29

**Authors:** Rabab Mohammed Bedawi, Yahya Al-Farsi, Hassan Mirza, Salim Al-Huseini, Tamadhir Al-Mahrouqi, Omaima Al-Kiyumi, Mohammed Al-Azri, Samir Al-Adawi

**Affiliations:** 1Department of Family Medicine and Public Health, College of Medicine and Health Sciences, Sultan Qaboos University, Muscat 123, Oman; rababbedawi1@gmail.com (R.M.B.); ymfarsi@squ.edu.om (Y.A.-F.); mhalazri@squ.edu.om (M.A.-A.); 2Department of Behavioural Medicine, College of Medicine and Health Sciences, Sultan Qaboos University, Muscat 123, Oman; mirza@squ.edu.om; 3Department of Psychiatry, Al Masarra Hospital, Ministry of Health, Muscat 113, Oman; salimkhalfan.al-huseini@nhs.net; 4Psychiatry Residency Program, Oman Medical Specialty Board, Muscat 130, Omanr20125@resident.omsb.org (O.A.-K.)

**Keywords:** adult ADHD, standardised prevalence estimates, tertiary hospital, age and sex differences, regional variations, subtype and comorbidities, prescription patterns, temporal trends

## Abstract

(1) Objectives: This study aimed to assess the 5-year prevalence and clinical profile of attention deficit hyperactive disorder (ADHD) among adult patients seeking care in a tertiary care hospital in Oman. (2) Methods: The data were analysed using descriptive and inferential statistics and standardised prevalence estimates were calculated. (3) Results: Of the 39,881 hospital visits, 1.77% were made by adults with ADHD. This is equivalent to 17.8 visits per 1000 outpatients. The year 2021 saw the highest prevalence among the five years considered, while 2020 had the lowest prevalence. Although the age distribution indicated that the age group ‘under 20’ had the highest prevalence, the gender distribution showed that ADHD was more common among adult men. Among the various subtypes of ADHD, inattention was the most common. (4) Conclusions: This study specifically compared the prevalence and associated factors between an adult cohort with ADHD and those other psychiatric clinic attendees during the same period. The study offers important information on the prevalence and clinical profile of adults with ADHD in the population under consideration.

## 1. Introduction

As countries in the global south, specifically those of the Arabian Gulf, experience an increase in their standard of living, new challenges have emerged, including a rising tide of non-communicable diseases such as attention deficit hyperactivity disorder (ADHD) [1]. Although typically considered a condition that affects children, studies have indicated that 30% to 50% of people who were diagnosed with ADHD during childhood continue to experience the primary symptoms of the disorder even after turning 18 years [2,3]. In recent times, there has been a growing interest in adult ADHD in countries of the Arab Gulf. However, the available data are limited to university students, and the presence of ADHD has been based mainly on ‘symptom checklists’ [4,5]. It is crucial to note that the structured questionnaires used in these studies can only identify subclinical symptoms of ADHD and not the entire spectrum of the disorder. Therefore, more research is required to establish a comprehensive understanding of ADHD among adults in Arab Gulf countries. Currently, there is a lack of studies conducted in hospital settings that could offer a carefully managed and regulated environment. Such an environment would facilitate access to a team of experienced healthcare professionals trained to diagnose the presence of ADHD. Additionally, this type of setting would be ideal for exploring possible comorbidity and other relevant clinical risk factors associated with ADHD.

Along with an exploration of the number of clinic visits for adult ADHD, studies are also needed to explore regional variation, age, sex differences, subtypes and comorbidities, and prescription patterns. Previous studies have explored the magnitude of adult ADHD through the lens of the urban–rural dichotomy [6]. Exploring the urban–rural dichotomy has the potential to uncover important data on regional variations [7]. This has the potential to shed light on disparities in access to services. Understanding regional variation can help policymakers and healthcare providers identify areas lacking resources and support and take appropriate measures to address discrepancies. To date, a regional variation has received little attention in the Arabian Gulf countries.

In addition to regional variations, studies are needed in Arab Gulf countries to explore gender differences between adult ADHD attendees seeking consultation in a tertiary care setting. Studies in other populations have estimated that around 4.4% of the population has ADHD, with a higher incidence among men than among women [8] and a male-female ratio of approximately 3:1 in clinical samples [9]. There is a gender-specific presentation of ADHD. Studies in other populations have indicated that ADHD in adults could manifest itself as an inattention, hyperactive-impulsive, or mixed subtype, while women predominantly exhibit an inattention subtype of ADHD [9]. Males exhibited a higher prevalence of ADHD than females, indicating possible gender-related variations in the presentation and diagnosis of the disorder. There are limited data on healthcare care utilisation and sex among adults with ADHD in Arab Gulf countries. Previous studies in the Arabian Gulf have suggested that, in terms of care seekers with poor mental health outcomes, there are gender differences in the utilisation of mental health services in Arabic countries, and women are potentially more likely to seek psychiatric inpatient care [10]. It remains to be established whether there is a gender-specific presentation of ADHD in hospital settings.

ADHD is commonly diagnosed in childhood. However, research indicates that it can persist into adolescence and adulthood [11]. ADHD symptoms often manifest differently in adults compared to children and adolescents [12]. Despite this knowledge, there is still ambiguity surrounding the prevalence of ADHD in adults, especially in different age groups. Therefore, studies are needed to examine the prevalence of ADHD in adulthood in different age groups.

Insufficient attention has been paid to the prescription patterns of medications for adults with ADHD in countries of the Arabian Gulf. In the global south, access to mental health services may be limited, including diagnosis and treatment of ADHD, compared to the global north [13]. It has not yet been determined whether cultural beliefs and norms can influence how ADHD is perceived, leading to potential variations in prescription patterns. Due to the scarcity of information available on prescription patterns for adults with ADHD, research is necessary to address this gap.

The discussion aforementioned appears to suggest that adult ADHD has been reported in the Arab Gulf population, but it is limited to community surveys [4,5]. Additionally, little information has been provided on characteristics such as urban–rural residency, gender, age, and prescription patterns among adults with ADHD attending a tertiary care hospital. To fill the literature gap, this study compares the prevalence and associated factors between a cohort with ADHD and those without it, visiting a tertiary hospital at the same time in Oman. The study investigated regional variations in ADHD rates in conjunction with an examination of how ADHD varied between age groups and between sexes. The current study also explored different subtypes of ADHD and their co-occurrence with other conditions. Furthermore, the study was designed to analyse prescription drug patterns for the treatment of ADHD.

## 2. Methods

### 2.1. Study Design

The study included a retrospective examination of reports detailing patient information about ADHD for 5 years, from January 2018 to 2022, at Sultan Qaboos University Hospital (SQUH).

### 2.2. Study Setting

In Oman, healthcare is provided by the Ministry of Health through a three-tier system comprising primary health centres, regional referral hospitals (offering secondary care healthcare), and specialised hospitals (providing tertiary care). This study was carried out at SQUH, Department of Behavioural Medicine, a 570-bed tertiary care facility located in Muscat, Oman. Within the three-tier healthcare system of Oman, SQUH functions as a teaching hospital and a referral centre serving the entire country. Established in 2018 as a pilot project, the Adult ADHD Clinic at SQUH became the only national centre for adult ADHD. This clinic addressed the needs of existing young patients with ADHD who were transitioning to adulthood, in addition to admitting new cases of adult ADHD [14].

### 2.3. Case Definition and Ascertainment

Routine procedures for patients with ADHD seeking consultation from the current adult ADHD clinic are as follows. The initial interview involved symptom review, and history-taking. This is followed by evaluations using rating scales or checklists (e.g., Conners’ Adult ADHD Rating Scales or the Adult ADHD Self-Report Scale-V1.1) [15,16].

Given the high comorbidity in adults with ADHD [17], evaluations of adult ADHD are often performed separately from acute episodes to ensure precision. This also involved a medical examination. If the respondent appears to have met the initial screening for adult ADHD, then he/she participates in the semi-structured interview using the style and format of the Composite International Diagnostic Interview (CIDI) for which the senior clinical member is trained. The CIDI has been developed to provide practitioners with a reliable and valid assessment of specific and general psychological disorders according to the definitions and criteria of the International Classification of Diseases (ICD-10) and the Diagnostic and Statistical Manual of Mental Disorders (DSM).

To delineate the different subtypes of ADHD, medical notes were carefully deciphered to determine whether the case is marked by one of the three subtypes of ADHD: (i) primarily hyperactive and impulsive, (ii) primarily inattentive and (ii) combined. Each of these subtypes is distinguished by a set of clinical symptoms in diagnostic criteria [18].

### 2.4. Study Participants

The study cohort included people with ADHD who sought treatment at the Adult ADHD Clinic during the study period. To account for the specific prevalence of diagnostics rather than just the prevalence of visits, which could reflect help-seeking behaviour rather than the actual prevalence of ADHD in the clinic, we also included all patients seeking consultation as part of the larger psychiatric department. The inclusion criteria involved adults— between 18 and 60 years of age. The exclusion criteria comprised a diagnosis of learning disorders, a IQ below 80, or findings with missing data. The study focused on SQUH in Oman to explore regional variations in the prevalence of adult ADHD. The regions were demarcated according to the pre-existing administrative regions of Oman, divided into 11 governorates (‘muhafazah’) [Figure 1]. The regions included Ad Dakhiliyah, Ad Dhahirah, Al Batinah, Al Buraymi, Al Wusta, Ash Sharqiyah, Dhofar, Muscat and Musandam, resulting in nine governorates for this study.

### 2.5. Sampling and Sample Size

To determine the sample size necessary for this study, we used Open Epi. We decided on an error rate of 2.5%, a significance level (type 1 error) of 5%, and a 95% confidence interval with a pre-existing estimate of 5% prevalence of adult ADHD among attendees of the Department of Behavioural Medicine and Psychiatry. The objective of the power analysis was to ensure that there was an adequate sample size based on a conservative a priori assumption of the prevalence of ADHD that was estimated based on the relevant reviewed literature. The calculation indicated that a sample size of 210 would be required to achieve a power of 80% in this study. Convenience sampling was used to recruit participants for this study.

### 2.6. Data Extraction

This study used data from the Hospital Information System (HIS), a secure centralised database that contained patient health information. The required information was extracted and stored in an Excel file and later converted to the Social Sciences Statistical Package (SPSS), version 24.0.

### 2.7. Data Management

Data were organised using an Excel sheet and included sociodemographic and clinical details of adult patients with ADHD, such as age, sex, marital status, education, occupation, clinical presentation, family history, as well as a history of substance abuse and/or the presence of any other psychiatric comorbidity. Furthermore, information on admission history, forensic history, and medications was collected. Data quality was ensured through an extensive cleaning process, including removing duplicates, standardising data, formatting, translating data when necessary, and correcting typographical errors.

### 2.8. Data Analysis

Statistical analysis involved describing, distributing and categorising the study variables. The chi-square test examined the relationships between categorical variables, with a significance set at 5%. The prevalence estimates of the clinic visits for adult ADHD were calculated by dividing the ADHD visits by the total outpatient visits of adults, reported per 1000 visits. Prevalence estimates and 95% confidence intervals (95% CI) were determined using the Poisson distribution, calculated using the GraphPad Prism 6.0 software. Stratification was performed by age, sex, residence, nationality, and level of education. Standardised governorate-specific prevalence estimates used indirect standardisation based on Oman’s 2020 census data. The distribution and categorisation of sociodemographic and clinical characteristics were performed among selected cases of adults with ADHD, as well as their history indices.

Statistical analysis was performed using SPSS (version 24.0, IBM) and Excel software-2020).

## 3. Results

### 3.1. Sociodemographic Characteristics of Adult Participants with Diagnosis of ADHD

Table 1 shows the sociodemographic characteristics of patients with ADHD who visited the Department of Behavioural Medicine and Psychiatry clinic during a 5-year follow-up period (2018–2022). Between 2018 and 2022, the Department of Behavioural Medicine and Psychiatry Clinic recorded 39,881 adult visits, with 709 (1.77%) visits (1.77%) of adults diagnosed with ADHD. For non-ADHD visits, the gender ratio was almost equal, but for ADHD visits, men made up nearly 70% (491 visits) and women just over 30% (218 visits). Most of the visitors were over 40 years old, but only 3.5% were between the ages of 18 and 19. However, among the ADHD group, 80% were under 30, with the early 20s group having the highest percentage of visits at 39.1%, and those in their late 30s the fewest at 3.4%. The visitors came from nine governorates, Muscat contributing to 45% of all visits and Al Batinah to 21.5%. Musandam, Dhofar and Al Wusta each had less than 1% of the total visits. The pattern of geographic origin was similar between general visitors and those to the ADHD clinic, with more than half (51.3%) from Muscat and less than 3% combined from Dhofar, Al Buraymi, Ad Dhahirah and Wusta. In particular, no ADHD patients came from the Wusta governorate.

### 3.2. Hospital Characteristics of Adults with a Diagnosis of ADHD

The results shown in Table 2 are presented as a comparison of the hospital characteristics of adult patients with ADHD with those without ADHD. In both the general department and the ADHD clinic, most adult patients were treated as outpatients, 97.9% and 98.3%, respectively, and only a small number were hospitalised, 2% in the general department and 1.7% in the ADHD clinic. The trend in visits from 2018 to 2022 for both the Department of Behavioural Medicine and Psychiatry and the ADHD clinic showed a peak in 2021 for the ADHD clinic with 23.7% of visits and in 2018 for the general department with 22.4%. The year 2020 had the lowest visitation rates for both settings, with a decrease of 15% in the ADHD clinic and 11.3% in the Department of Behavioural Medicine and Psychiatry.

The distribution of ADHD visits between different age and sex groups in the Department of Behavioural Medicine and Psychiatry during five years of follow-up is presented in Table 3 and Figure 2. Over five years, the Department of Behavioural Medicine and Psychiatry recorded ADHD visits, with a focus on different age and sex groups. The younger age group had the highest number of visits from adults with ADHD, which decreased with the age of the patients. The highest prevalence of visits with ADHD was approximately 105 per 1000 visits among those under 19 years of age. The lowest prevalence of visits was about 2 per 1000 visits in the oldest age group.

Gender-specific prevalence estimates for different age categories are shown in Figure 3. In all age groups, except for those over 40 years old, men were more likely to visit the Adult ADHD Clinic than women. Men in their early 20s and older adolescents over 19 years of age had the highest visit rates to the ADHD clinic, with approximately 125 and 179 per 1000 visits, respectively. The lowest prevalence estimates among men were observed in those over 40 years of age and in their late 30s, with approximately 1 and 7 per 1000 visits, respectively. Women over 40 had the lowest prevalence estimates among all age groups of women, with approximately 2 per 1000 visits.

### 3.3. Governorate-Specific Visit Prevalence Estimates of Adult Attendees with Diagnosis of ADHD

The prevalence of ADHD visits in various governorates of Oman over five years is presented in Table 4. Over five years, the study tracked ADHD visits across Oman’s governorates, finding that Dhofar had the highest prevalence at 36.9, with a confidence range of 20.2 to 61.9. The prevalence of Dhofar was almost the sum of the prevalences in Muscat (20.3) and A’ Dakhiliyah (16.2) combined. A’ Dhahirah had the lowest prevalence of ADHD visits at 1.8, with a confidence range from 0.6 to 4.4. Al Wusta did not have any recorded prevalence of ADHD visits. Figure 4 visually compares these standardised prevalence estimates across governorates.

### 3.4. Prevalence of Visits to Adult ADHD by Year of Visit

Table 5 presents the prevalence estimates of ADHD visits by year of visit during the study period of five years in Oman. In the first year, there were approximately 17 ADHD visits per 1000, with a range from 14.1 to 19.4. In 2019, the prevalence increased slightly to 18.4 visits per 1000, ranging from 15.7 to 21.4. In 2020, there was a significant drop in prevalence to 13.3 visits per 1000, with a range of 10.7 to 16.5. The prevalence pattern fluctuated over the next two years, increasing to 21.7 visits per 1000 (with a range of 18.6 to 25.1) in 2021 and then dropping to 18.0 visits per 1000 (with a range of 15.4 to 21.0) in 2022.

### 3.5. Sociodemographic Characteristics of Selected Cases of Adults with ADHD

Table 6 shows the sociodemographic characteristics of adults with ADHD who followed up in the Department of Behavioural Medicine and Psychiatry for 5 years. The study involved 193 participants with a male-to-female ratio of 1.5:1. Most of the participants were unmarried. Most of the students had completed high school. Around one-third of the participants were in their 20s. Most of the participants were students or unemployed. Male and female participants had similar age distributions. More women were married and divorced than men. Women were more likely to have a bachelor’s or master’s degree, while only one man held a Ph.D. A higher percentage of men were students and more women were unemployed.

### 3.6. Subtypes of Adults with ADHD: Comorbidities

Table 7 shows selected clinical characteristics, including the subtypes observed in adults with ADHD, comorbidities, and medications. The centre follows the ICD-10 guidelines for ADHD classification. The inattention subtype was the most common form of ADHD, representing 63.5% of cases, with similar proportions between sexes. The hyperactive subtype was rare, found in 13.7% of women and 8.3% of men. Substance misuse was the leading comorbidity in patients with ADHD, with 32.4% experiencing drug abuse. This was higher in men (over 37%) than in women (over 23%). Anxiety was the second most common comorbidity, affecting more than 20% of ADHD patients, more commonly in women (25.9%) than in men (19.8%). Depression and personality disorders affected 15.1% of adults with ADHD, and both conditions were more prevalent in women. About 20% of women had depression or a personality disorder, compared to approximately one-seventh of men for depression and one-ninth for personality disorder. Women had almost twice the rate of personality disorders compared to men (20.7% vs. 11.2%). Suicide attempts were reported in 4.5% of patients with ADHD, with a slightly higher percentage in women (5.2%) than in men (4.3%). Psychosis was rare, found only in 1.7% of men, and was not reported in women. Methylphenidate was the most prescribed medication for ADHD, administered to more than three-quarters of patients. Atomoxetine was the second most common, used by 16.5% of the patients, with more women (20.7%) prescribing it than men (13.8%).

### 3.7. Selected Clinical History Indices among Selected Adult Attendees with a Diagnosis of ADHD

The results presented in Table 8 indicate the gender-specific clinical history of adult patients with ADHD. Of the total, 27.2% of patients in the Department of Behavioural Medicine and Psychiatry had at least one family member with ADHD. A higher percentage of men (31.1%) than women (21.9%) reported having a family history of ADHD. A history of hospital admission was observed in a minority of patients, 10.9% of whom were women and 5.6% being men. Only 2.2% of all patients had criminal records and men were more likely to have them than women. Only 15.5% of the patients had a history of bipolar affective disorder, with a slightly higher incidence in women than in men.

## 4. Discussion

Against the backdrop of existing research gaps in the global south, particularly in the Arabian Gulf, the current study aimed to explore the various dimensions of adult ADHD in Oman with a specific focus on understanding its prevalence, temporal trends, regional variances, age and sex differences, subtypes and comorbidities, and prescription patterns. The data indicated that adults with ADHD represented 1.77% of the 39,881 hospital visits, and therefore the prevalence was 17.8 per 1000 outpatient visits. There is regional variation, but the number of visits is skewed towards those who live near the urban and national capital, Muscat. The data indicated that 2021 saw more outpatient visits from adults with ADHD for the period under scrutiny here. The age group “under 20” appeared to have the most outpatient visits. In terms of sex, there was a preponderance of adult men and subtypes of inattention. Methylphenidate was the most commonly prescribed medication. These results are recapitulated below within the existing literature.

Although prevalence rates for adult ADHD show variability between studies and regions, according to a global systematic review and meta-analysis [19], the prevalence of persistent adult ADHD appears to constitute 2.58% of the sample surveyed, and that of symptomatic adult ADHD was 6.76%. Another study that focused on populations from different countries is the WHO World Mental Health Survey; Fayyad et al. [20] have used the CIDI tool, as in the present study, the prevalence of adult ADHD was found to constitute 2.8% of the sample (*n* = 26,744). In a systematic review and meta-analysis that covered the region of the Middle East and North Africa, a prevalence of 13.5 was observed [21].

The present appears to differ from the global trend. The current study suggests 1.77% outpatient visits, which represents a lower figure compared to the international trend. It is worth noting that the figure is described among hospital attendees, whereas previous studies have largely focused on community surveys. Other factors that could explain the divergent rates of adult ADHD include different screening tools, heterogeneous data collection methods, and reporting standards. Most epidemiological surveys in the global south have relied on ‘symptom checklists’ [22]. In the present setting, the diagnosis of ADHD adhered to the gold standard interview based on ICD and DSM. Therefore, more studies are needed to take into account the divergent prevalence rate.

In addition, to examine the frequency of outpatient visits, this study also examines various social demographics and risk factors for adults with ADHD. A significant portion of the participants self-identified as single. This observation aligns with existing research that highlights a higher likelihood that relationships end when one partner is affected by ADHD, compared to cases where ADHD is not present in the relationship [23]. The implications of ADHD on relationship dynamics are likely to be complex and multifaceted. Individuals with ADHD are easily distracted, have poor temporary organisation of behaviour, and have poor self-regulation. Such dispositions may be likely to negatively impact romantic or marital relationships. In another study, de Zwaan et al. [24] have reported that adults diagnosed with ADHD tend to have lower levels of marital satisfaction compared to their partners.

A considerable proportion of the male participants had high school diplomas, which is in line with the academic challenges often associated with ADHD [25]. On the contrary, a notable percentage of women had bachelor’s or master’s degrees. In Oman, there has been notable integration of women into spheres that were historically reserved for men. This empowerment of women appears to also expand in the tertiary education and employment sectors [26]. More research is needed to explore why women excel despite having ADHD, which appears to have hampered men’s progress in education.

The present study aims to explore which subtype of ADHD is more common in Oman. In the Italian population (Milan), Salvi et al. [27] reported that the inattentive subtype comprised 18.3% of their sample, while the hyperactive/impulsive subtype and the combined subtype comprised 8.3% and 70% of the sample, respectively. On the contrary, this study suggests that most patients exhibited the subtype of inattention. Some studies have indicated that there is a gender difference in the subtypes of ADHD in adults. However, in these Oman data, despite the prominence of the inattention subtype, no significant gender-based differences emerged, a trend that echoes previous research that highlights the commonality of this subtype in adults [28].

This study also explored prescription patterns among adults receiving ADHD treatment. The medications commonly prescribed for adults with ADHD are often categorised as stimulants or non-stimulants. In the present centre, methylphenidate (Stimulant type) was the most prescribed medication (over 75% of patients), consistent with current guidelines [29]. Atomoxetine, often labelled a non-stimulant, was the second most prescribed drug (16.5% of patients), with a higher proportion of women receiving it compared to men. Other medications were prescribed less frequently. As noted above, the present cohort appeared to have many comorbidities. Adults with ADHD have been reported to use various classes of psychiatric medications, such as antidepressants, antipsychotics, anxiolytics, mood stabilisers, sedatives, and antidepressant-enhancing agents [30]. As reported in the existing literature, although prescribed for adults with ADHD, the use of these compounds is not widespread. However, what appears to be conspicuously absent, particularly in studies in the global south, is non-pharmacological treatment despite the available evidence for the efficacy of non-pharmacological agents [31].

### 4.1. Strengths

One of the strengths of this study is that the study took a ‘specific diagnostic prevalence’ approach rather than ‘visit prevalence’ [32]. In the current study, the specific diagnostic prevalence has been defined as the proportion of adults with ADHD who seek consultation for a specific period (2018–2022). For better consequentiality, this study extracted diagnostic prevalence rates in the context of general visit prevalence rates by calculating the number of people with ADHD out of the total number of people with other psychiatric conditions attending SQUH. Second, unlike previous studies from the global south that have reported the magnitude of ADHD using symptom checklists, this study has sought prevalence estimates using clinically defined diagnostic criteria [26].

### 4.2. Limitations

Despite the valuable information offered by this study on adult ADHD, certain limitations merit consideration in the interpretation of the results. First, relying on data from a single adult ADHD clinic may not fully represent the larger population of adults with ADHD in Oman, as those seeking treatment in this specific clinic may differ from those seeking treatment elsewhere or not seeking treatment at all. Second, reliance on hospital records can result in incomplete or inaccurate information, lacking certain details about the medical history of patients or the reasons behind trends. Prospective studies are recommended in this regard. Third, the study does not adequately explore external factors that influence department visits, such as social attitudes, awareness campaigns, or policy changes that affect healthcare access. Fourth, the limitation of our study is that to perform an accurate diagnosis in adulthood, specific tools are necessary, such as diagnostic interviews, self-tests, informant reports, semi-structured clinical interviews, and medical history. The diagnosis in adults is retrospective, so the presence of symptoms in childhood, before the age of 12, should have been considered a criterion. This is one of the limitations of this study. Lastly, there may be a self-selection bias, where clinic visitors differ from non-visitors in symptom severity, healthcare access, or willingness to seek treatment, impacting sample representativeness. To address these limitations and improve the robustness of the study, future research could employ prospective and longitudinal designs, combining quantitative and qualitative methods for a more complete understanding of adult ADHD and its associated factors.

## 5. Conclusions

The take-home message of this study is that ADHD is not only a childhood disease but also affects adults, with a prevalence of 1.77% among hospital visits in Oman. The year 2021 saw the highest prevalence, while 2020 had the lowest. ADHD was more common among adult men and the inattention subtype was the most frequent. This study highlights the importance of exploring covariates of adults with ADHD attending healthcare settings, including regional sociodemographic factors, clinical risk factors, regional variations, and prescription drug patterns for the treatment of ADHD. Meanwhile, these variables could be used to prevent and mitigate the challenges faced by adult ADHD.

## Figures and Tables

**Figure 1 ijerph-21-00566-f001:**
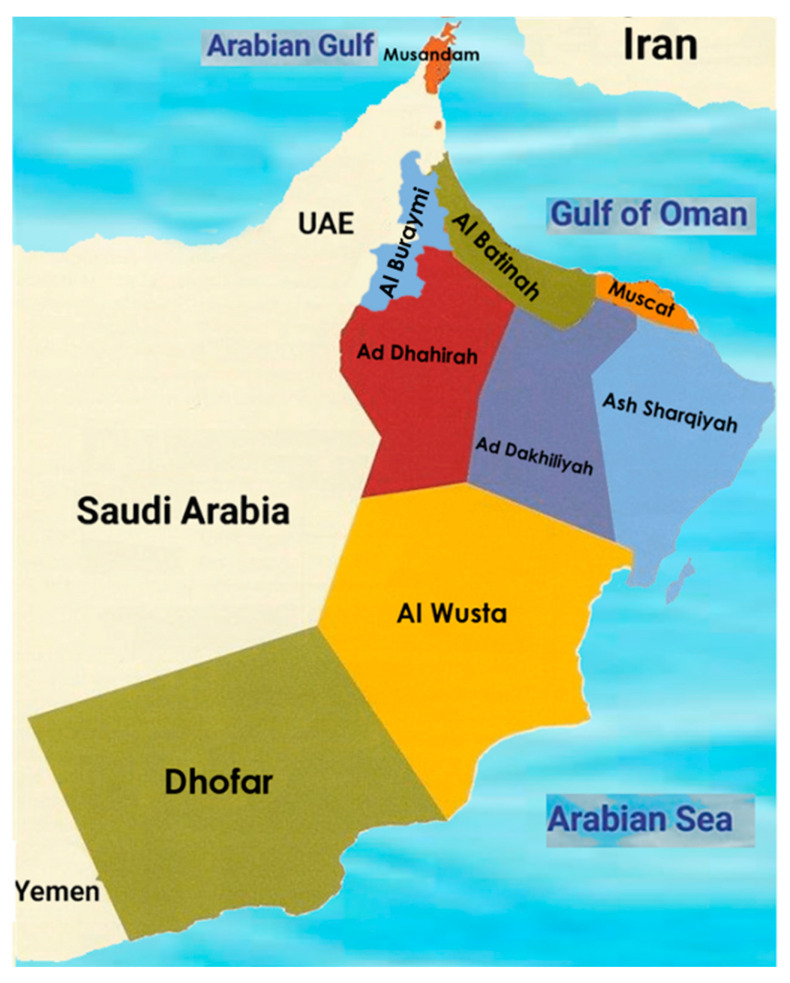
Map of the Sultanate of Oman and its governorates. Adapted from the Anglo-Omani Society (https://images.app.goo.gl/1fDC8xuouzq2T3MB7, accessed on 20 February 2024).

**Figure 2 ijerph-21-00566-f002:**
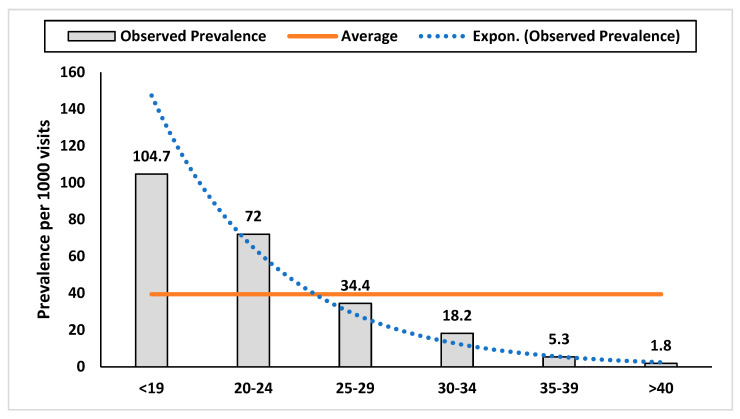
Distribution of prevalence estimates (per 1000 visits) of visits to adults diagnosed with ADHD by age category, Oman, 2018–2022.

**Figure 3 ijerph-21-00566-f003:**
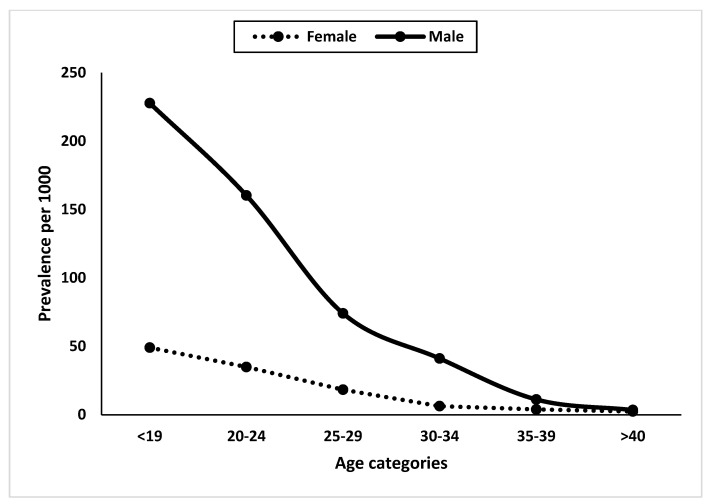
Distribution of the gender-specific prevalence (per 1000 visits) of adult attendees with the diagnosis of ADHD by age category.

**Figure 4 ijerph-21-00566-f004:**
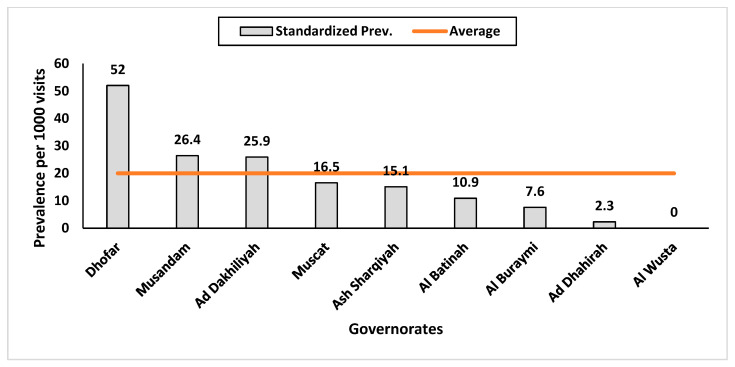
Governorate-specific estimates of the prevalence of standardised visits of adult attendees with the diagnosis of ADHD among all adult visits to the Department of Behavioural Medicine and Psychiatry over five years of follow-up, Oman, 2018–2022.

**Table 1 ijerph-21-00566-t001:** Sociodemographic characteristics of all adult visits to the Department of Behavioural Medicine and Psychiatry during five years of follow-up, Oman, 2018–2022.

Characteristics	Total	ADHD Visits	Non-ADHD Visits	*p*-Value
	(N = 39,881) N (%)	(N = 709) N (%)	(N = 39,172) N (%)	
Gender				0.02 *
Female	23,089 (57.9)	218 (30.7)	22,871 (58.4)	
Male	16,792 (42.1)	491 (69.3)	16,301 (41.6)	
Age (in years)				0.001 *
19 or less	1394 (3.5)	146 (20.6)	1248 (3.2)	
20 to 24	3845 (9.6)	277 (39.1)	3568 (9.1)	
25 to 29	4304 (10.8)	148 (20.9)	4156 (10.60	
30 to 34	4060 (10.2)	74 (10.4)	3986 (10.2)	
35 to 39	4495 (11.3)	24 (3.4)	4471 (11.4)	
40 or above.	21,783 (54.6)	40 (5.6)	21,743 (55.5)	
Governorate				0.001 *
Muscat	17,917 (44.9)	364 (51.3)	17,553 (44.8)	
Al Batinah	8562 (21.5)	129 (18.2)	8433 (21.5)	
A’ Dakhiliyah	6978 (17.5)	113 (15.9)	6865 (17.5)	
A’ Sharqiyah	3328 (8.3)	44 (6.2)	3284 (8.4)	
Musandam	130 (0.3)	38 (5.4)	92 (0.2)	
Dhofar	325 (0.8)	12 (1.7)	313 (0.8)	
Al Buraymi	415 (1)	5 (0.7)	410 (1)	
A’ Dhahirah	2167 (4)	4 (0.6)	2163 (5.5)	
Al Wusta	59 (0.1)	0	59 (0.2)	

ADHD: Attention deficit hyperactivity disorder. * Significant *p*-values at *p* < 0.05.

**Table 2 ijerph-21-00566-t002:** Characteristics of hospital inpatient and outpatient visits of adult participants in the Department of Behavioural Medicine and Psychiatry during five years of follow-up, Oman, 2018–2022.

Characteristics	Total	ADHD Visits	Non-ADHD	*p*-Value
	(N = 39,881) N (%)	(N = 709) N (%)	(N = 39,172) N (%)	
Visit type				0.28
Inpatient	838 (2.1)	12 (1.7)	826 (2.1)	
Outpatient	39,043 (97.9)	697 (98.3)	38,346 (97.9)	
Year of visit				0.07
2018	8939 (22.4)	148 (20.9)	8791 (22.4)	
2019	8545 (21.4)	157 (22.1)	8388 (21.4)	
2020	5997 (15.0)	80 (11.3)	5917 (15.1)	
2021	7754 (19.4)	168 (23.7)	7586 (19.4)	
2022	8646 (21.7)	156 (22.0)	8490 (21.7)	

Significant *p*-values at *p* < 0.05. Estimates of the prevalence of age and sex-specific visits of adult attendees with a diagnosis of ADHD.

**Table 3 ijerph-21-00566-t003:** Estimates of the prevalence (per 1000 visits) of age and sex-specific visits among adult attendees with a diagnosis of ADHD.

Age	Gender	Total Visits	Visits to the ADHD Clinic	P (95% CI)
		N	N	
Overall	Total	39,881	709	17.8 (16.5–19.1)
	Female	23,089	218	9.4 (8.3–10.8)
	Male	16,792	491	29.2 (26.8–31.9)
19 or less	Total	1394	146	104.7 (89.5–121.6)
	Female	795	39	49.1 (35.6–65.8)
	Male	599	107	178.6 (149.5–210.9)
20 to 24	Total	3845	277	72.0 (64.2–80.5)
	Female	2266	79	34.9 (27.9–43.0)
	Male	1579	198	125.4 (109.7–142.4)
25 to 29	Total	4304	148	34.4 (29.2–40.2)
	Female	2457	45	18.3 (13.6–24.2)
	Male	1847	103	55.8 (46.0–66.9)
30 to 34	Total	4060	74	18.2 (14.4–22.7)
	Female	2364	15	6.3 (3.7–10.2)
	Male	1696	59	34.8 (26.8–44.3)
35 to 39	Total	4495	24	5.3 (3.5–7.8)
	Female	2586	10	3.9 (2.0–6.9)
	Male	1909	14	7.3 (4.2–12.0)
40 or above.	Total	21,783	40	1.8 (1.3–2.5)
	Female	12,621	30	2.4 (1.6–3.3)
	Male	9162	10	1.1 (0.6–1.9)

**Table 4 ijerph-21-00566-t004:** Governorate-specific estimates of the observed and standardised prevalence (per 1000 visits) of adult attendance visits with the diagnosis of ADHD among all adult visits to the tertiary care unit over five years of follow-up, Oman, 2018–2022.

Governorate	Total Visits	Visits to the ADHD Clinic	Observed Prevalence P (95% CI)	Standardised Prevalence P (95% CI)
	N	N		
Dhofar	325	12	36.9 (20.2–61.9)	52.0 (28.7–84.7)
Musandam	130	21	16.5 (13.7–19.7)	26.4 (22.4–30.6)
A’ Dakhiliyah	6978	116	16.2 (13.4–19.4)	25.9 (21.9–30.2)
Muscat	17,917	373	20.3 (18.3–22.5)	16.5 (14.7–18.4)
A’ Sharqiyah	3328	45	13.2 (9.7–17.5)	15.1 (11.2–19.9)
Al Batinah	8562	132	15.1 (12.6–17.8)	10.9 (8.9–13.0)
Al Buraymi	415	5	12 (4.4–26.5)	7.6 (2.4–18.2)
A’ Dhahirah	2167	4	1.8 (0.6–4.4)	2.3 (0.8–5.5)
Al Wusta	59	0	0	0

**Table 5 ijerph-21-00566-t005:** Estimates of the prevalence (per 1000 visits) of visits by adults with ADHD between 2018 and 2022.

Year of Visit	Total Visits	Visits to ADHD Visits	Observed Prevalence P (95% CI)
	N	N	
2018	8939	148	16.6 (14.1–19.4)
2019	8545	157	18.4 (15.7–21.4)
2020	5997	80	13.3 (13.3–16.5)
2021	7754	168	21.7 (21.7–25.1)
2022	8646	156	18.0 (18.0–21.0)

**Table 6 ijerph-21-00566-t006:** Sociodemographic characteristics of selected cases of adults with ADHD between 2018 and 2022 at Sultan Qaboos University Hospital in Oman, stratified by sex (N = 193).

Characteristics	Total	Female	Male	*p*-Value
	(N = 193) N (%)	(N = 77) N (%)	(N = 116) N (%)	
Marital status				0.10
Single	156 (80.7)	58 (74.7)	99 (84.9)	
Married	32 (16.6)	15 (20)	16 (14.2)	
Divorced	5 (2.7)	4 (5.3)	1 (0.9)	
Educational level				0.11
No high school diploma	26 (13.6)	8 (10.0)	18 (15.9)	
High School Diploma	84 (43.5)	28 (36.7)	56 (47.9)	
Bachelor’s degree	36 (18.8)	17 (21.6)	20 (17.0)	
MSc	45 (23.4)	24 (31.7)	21 (18.1)	
Ph.D.	2 (0.7)	0	1 (1.1)	
Age group				0.99
18–19	36 (18.4)	14 (18.4)	21 (18.4)	
20–24	64 (33.2)	25 (31.6)	39 (34.2)	
25–29	51 (26.2)	21 (27.6)	29 (25.4)	
30–34	21 (11.1)	8 (10.5)	13 (11.4)	
35 or above	21 (11.1)	9 (11.8)	12 (10.5)	
Occupational status				0.26
Governmental employee	21 (10.9)	6 (8.3)	15 (12.6)	
Private employee	49 (25.1)	21 (27.8)	27 (23.3)	
Unemployed	45 (23.4)	23 (29.2)	22 (19.4)	
Student	78 (40.6)	27 (34.7)	52 (44.7)	

Significant *p*-values at *p* < 0.05.

**Table 7 ijerph-21-00566-t007:** Selected clinical characteristics among adults with ADHD stratified by sex in the tertiary care unit, Oman, 2018–2022.

Characteristics	Total	Female	Male	*p*-Value
	(N = 193) N (%)	(N = 77) N (%)	(N = 116) N (%)	
Primary Clinical Presentation				0.16
Hyperactivity	20 (10.5)	10 (13.7)	10 (8.3)	
Inattentive	123 (63.5)	52 (67.1)	71 (61.1)	
Mixed	50 (26)	15 (19.2)	35 (30.6)	
Secondary clinical presentation (comorbidity)				
Depression	29 (15.1)	14 (18.3)	16 (13.8)	0.21
Anxiety	43 (22.0)	20 (25.9)	23 (19.8)	0.20
Obsessive Compulsive Disorder	6 (3.1)	1 (1.3)	5 (4.3)	0.22
Learning disorders/scholastic skills problems	13 (6.7)	4 (5.2)	9 (7.8)	0.47
Suicide attempts	9 (4.5)	4 (5.2)	5 (4.3)	0.37
Personality disorder	29 (15.1)	16 (20.7)	13 (11.2)	0.02
Psychosis	2 (1.1)	0	2 (1.7)	0.35
Substance abuse	62 (32.4)	18 (23.4)	43 (37.1)	0.04
Medications				
Methylphenidate	147 (76.1)	59 (76.6)	88 (75.9)	0.45
Atomoxetine	32 (16.5)	16 (20.7)	16 (13.8)	0.15
* Other	42 (23.4)	17 (24.7)	25 (22.4)	0.82

OCD: obsessive compulsive disorder. * Other medications include paroxetine, escitalopram, lorazepam, sertraline, fluoxetine, imipramine, quetiapine, bupropion, mirtazapine, clonazepam, paliperidone, clonidine, guanfacine, lisdexamfetamine, lamotrigine, lithium, alprazolam.

**Table 8 ijerph-21-00566-t008:** Clinical history indices among selected adult participants with a diagnosis of ADHD, stratified by sex, Oman, 2018–2022.

Dimensions	Total	Female	Male	*p*-Value
	(N = 179) N (%)	(N = 73) N (%)	(N = 106) N (%)	
Family history of ADHD	49 (27.2)	16 (21.9)	33 (31.1)	0.12
Family history of bipolar affective disorder	28 (15.5)	12 (16.4)	16 (15.9)	0.46
History of hospital admissions	14 (7.8)	8 (10.9)	6 (5.6)	0.14
Forensic history	4 (2.2)	0	4 (3.7)	0.12

BAD: Bipolar affective disorder. Significant *p*-value at *p* < 0.05.

## Data Availability

Data supporting the findings of this article are available upon request from the corresponding author: adawi@squ.edu.om.

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
