# Peer review of "Prevalence and Clinical Profile of Adults with ADHD Attending a Tertiary Care Hospital for Five Years"

_ijerph, 2024, doi:10.3390/ijerph21050566_

Round 1
Reviewer 1 Report
Comments and Suggestions for Authors
Dear Authors,
Firstly, I would like to congratulate you on contributing to the dissemination of such an important topic as ADHD in adulthood with this research. Below are my thoughts on the methods used:
Among your strengths, you mention the diagnosis; however, to perform an accurate diagnosis in adulthood, specific tools are necessary, such as diagnostic interviews, self-tests, informant reports, semi-structured clinical interviews, and medical history. The diagnosis in adults is retrospective, so the presence of symptoms in childhood, before the age of 12, must be considered as a criterion.
In the article, I did not see mention of the tools used, except for the DSM, nor the method of conducting the diagnosis.Could you expand on the section regarding the method used for diagnosis?
In the introduction, you mention the transition phase, but in the sample selection, there is no mention of patients who arrived at your center with a previous diagnosis of ADHD or with suspicion of a first diagnosis. I was also wondering what criteria were used to select patients coming to the hospital and clinic for ADHD. Did you conduct screening on them? What test did you use?
Regarding the methods, there is a lack of information on the tests used to investigate comorbidities.
What instruments did you use, besides the DSM criteria?
Regarding the topic of transition into adulthood, I suggest the article "Transition care for adolescents and young adults with attention-deficit hyperactivity disorder (ADHD): A descriptive summary of qualitative evidence."
I hope these comments can be useful for your publication.
Best regards,
Author Response
Please see our responses to your comments in the enclosed point-counterpoint Table.

Reviewer 2 Report
Comments and Suggestions for Authors
The present study was designed to determine the periodic prevalence of ADHD in adults referring to a Tertiary Care Hospital during Five Years.
It seems that the objective of the current study, in addition to determining the prevalence of ADHD in adults, is also to determine the trends and changes in the pattern of clients to a tertiary clinic. Anyway, what is written as the overarching goal of the current study (to understand the universal aspects of adult ADHD and the cultural influences) is not reachable with this design.
The introduction of the manuscript is long and in many parts is not related to the purpose of the current study. It should be concise and focused on the main objective of the study.
The sample size required to determine prevalence does not require power. If the power is mentioned, it should be specified in line with which analytical goal it is.
In the first part of the discussion, the main results obtained in the present study should be mentioned.
Author Response
We appreciate REVIEWER 2's constructive critique of our manuscript, and we are pleased to report that we have addressed all the points raised. First, we have removed the statement regarding understanding universal aspects of adult ADHD and cultural influences. Secondly, we have significantly shortened and streamlined the introduction to ensure it aligns more closely with the study's main objective. Finally, we have clarified the power of determining sample size concerning specific analytical goals. We are grateful for these insightful comments and remain committed to improving the quality of our manuscript.

Reviewer 3 Report
Comments and Suggestions for Authors
Thank you for submitting your review “Prevalence and clinical profile of adults with ADHD attending a tertiary care hospital during five years: a cross-sectional study”
The manuscript provides interesting data on the prevalence of ADHD in Oman. It is nice to see data from another country than one classified as a high-income country.
My suggestions and feedback on the manuscript primary is focussed on being more concise. The introduction is long and covers content that is not related to project. Setting this aside, stating the aim in the middle of the introduction (third paragraph) segregated the introduction. The text thereafter details prevalence estimates in other countries to the point that made me question if the study was needed. The content could be more conscise and with this present a stronger rationale for the need of the study.
The overarching objective which is stated at the end of the introduction is not strongly connected to the aim stated above. The introduction would be stonger if these were aligned (if both the aim and objective end up being stated), and ending with the aim.
A similar thought transpires to the discussion. It could be more concise.
I have a few queries about the methods and results. The study design states the project is a retrospective examinations of reports but the methods are written as if it as prospective study. Could this please been clarified.
I suggest revising the tenses at times, for example, in the case definition section, the text is written in present tense.
The content in the data collection section merges details about data handing and data analyses. The manuscript could be improved by keeping such details separate.
In the results, some clarity is needed at
Line 219, what does “in general” mean?
Line 223 what is “significantly different” mean here?
Line226 what does “most” consultations mean?
Line 228 what is “when specifically looking at”. What does this mean?
In Section 3.4, the results are not clear representation of what was reported in the methods. It feels like a lot of post-hoc analyses. Were these planned a priori?
For the tables and figures
Could “years” be added to Table 1 re age
Table 2. Regarding data related to “year”, is this inpatient or outpatient. It is confusing when looking at the table in isolation and seeing the rows above.
Table 3 The details of the footnote should be labelled in the header.
Figure 2, what is the unit of analysis?
Overall, the results should be written as facts. There are a lot of words to describe the result that could be done more concisely. Much of the text rehashed the content in the tables.
A final comment, there are an exorbitant number of references for a manuscript that is not a systematic review with a large number of included studies. Are all those references included as in-text citations? If that many of references are used in a paper I question if the paper is needed. More isn’t always better.
Comments on the Quality of English LanguageThe use of English is fine. A minor check is needed to confirm correct use of tenses in the methods.
Author Response
The esteemed REVIEWER 3 has requested that we improve the conciseness throughout the manuscript, particularly in the introduction and discussion sections, to strengthen the rationale for the study. REVIEWER 3 has suggested that we align the aim and overarching objective, with a preference for ending the introduction with the aim. REVIEWER 3 suggested points for the methods and results: clarifications on study design, tense consistency, and separating data handling data analysis details. REVIEWER 3 recommended that attention be paid to the Tables and Figures. In addition, the review suggests reducing the number of references and ensuring correct tense usage throughout the manuscript.
As detailed at the point-counterpoint in the enclosed table, we have addressed the suggestions provided in REVIEWER 3. We appreciate the thorough feedback and believe the revisions have significantly enhanced the manuscript's quality. Should there be any additional points for improvement, we are eager to address them accordingly.

Round 2
Reviewer 3 Report
Comments and Suggestions for Authors
Dear Authors,
Thank you for updating the manuscript.
Two minor edits,
- There is a typo on line 415, where the first part of the sentence is missing.
- Figure 3 needs a y-axis label.
Comments on the Quality of English LanguageEnglish quality is ok. Some minor typos to be fixed on proof editing.
Author Response
Dear Colleague,
We would like to extend our sincerest gratitude for your thoughtful and constructive feedback on our manuscript titled " Prevalence and Clinical Profile of Adults with ADHD Attending a Tertiary Care Hospital for Five Years" Your invaluable in enhancing comments has helped to improve the quality and clarity of our submission. We have attended to all your suggestions. The corrections are now shown via the track change system.
